# Comparative Analysis of Five Moroccan Thyme Species: Insights into Chemical Composition, Antioxidant Potential, Anti-Enzymatic Properties, and Insecticidal Effects

**DOI:** 10.3390/plants14010116

**Published:** 2025-01-03

**Authors:** Mohamed Ouknin, Hassan Alahyane, Naima Ait Aabd, Sara Elgadi, Youssef Lghazi, Lhou Majidi

**Affiliations:** 1Regional Center of Agricultural Research of Agadir, National Institute of Agricultural Research (INRA), Avenue Ennasr, BP415 Rabat Principale, Rabat 10090, Morocco; naima.aitaabd@inra.ma; 2High Institute of Nursing Professions and Health Techniques, Beni Mellal 23000, Morocco; alahyanerh@gmail.com; 3Laboratory of Microbial Biotechnology, Agrosciences and Environment, Team of Agrosciences, PhytoBiodiversity and Environment, Faculty of Sciences Semlalia, Cadi Ayyad University, BP. 2390. 40000, Marrakech, Morocco; sarah.elgadi@gmail.com; 4Bio-Geosciences and Materials Engineering Laboratory, Higher Normal School, Hassan II University of Casablanca, Casablanca 20250, Morocco; 5Laboratory of Nanotechnology, Materials and Environment, Department of Chemistry, Faculty of Science, University Mohammed V, Rabat 10106, Morocco

**Keywords:** thyme species, essentials oils, chemical composition, biological activities, lethal concentrations, *Aphis gossypii*

## Abstract

This study aimed to investigate the chemical composition and bioactivities of essential oils (EOs) from five Moroccan thyme species: *Thymus broussonetii* subsp. *broussonetii, T. maroccanus*, *T. willdenowii*, *T. zygis* subsp. *gracilis*, and *T*. *satureioides*, collected from various geographical regions. Gas chromatography–mass spectrometry (GC-MS) identified thymol, *p*-cymene, borneol, γ-terpinene, carvacrol, α-pinene, and camphene as major constituents, with variations across species. Inductively coupled plasma atomic emission spectroscopy (ICP-AES) revealed important levels of calcium (450.6–712.2 mg/kg), potassium (255.4–420.7 mg/kg), magnesium (97.3–150.7 mg/kg), and iron (1.95–15.1 mg/kg). The EOs demonstrated strong antioxidant activity in DPPH, FRAP, and *β*-carotene bleaching assays. Insecticidal activity against *Aphis gossypii* revealed the highest efficacy with *T. willdenowii* EO (LC_50_ = 6.2 µL/mL), followed by *T. maroccanus* and *T. zygis* subsp. *gracilis*. Additionally, the EOs exhibited potent enzyme inhibitory effects at 1 mg/mL on acetylcholinesterase (83.1–96.4%), tyrosinase (77.5–89.6%), and α-glucosidase (79.4–89.5%). These findings suggest that thyme EOs, particularly from *T. willdenowii*, *T. zygis*, and *T. maroccanus*, are promising candidates for integrated pest management and natural enzyme inhibitors. Their potential applications in medicinal and pharmaceutical fields underscore the need for further research to optimize their use under specific conditions and dosages.

## 1. Introduction

Medicinal plants are rich in secondary metabolites, including essential oils (EOs), phenolic compounds, and flavonoids, which demonstrate a wide range of biological activities [1,2,3,4]. Famed for their therapeutic properties, these plants have been utilized in traditional medicine for over 2000 years in countries such as China, Greece, Egypt, and India [5]. In China, medicinal plants have been traditionally used to address various health conditions such as fever-inducing infections, the common cold, inflammatory disorders, and hepatitis [6]. Among these, the Lamiaceae family stands out, comprising over 236 genera, with *Thymus* being one of its most prominent genera.

*Thymus* L. encompasses over 214 species and 36 subspecies globally [7]. In Morocco, this genus stands out for its significant richness, diversity, and socio-economic importance, with around 21 species of thyme identified, 12 of which are endemic to the region [8]. These species are widely utilized in traditional medicine for their notable therapeutic properties. They are commonly prepared as infusions or decoctions and are used to treat various ailments, including dysentery, colitis, gastrointestinal disorders, and gastric acidity [9]. Moreover, several studies showed that the Moroccan *Thymus* species exhibits good anti-inflammatory, anticoagulant, and antioxidant activities [10,11]. Additionally, *Thymus* species are widely utilized in natural preservation, particularly in the food industry, owing to the complex composition of their extracts [12,13]. Among these, essential oils are the most commonly utilized form, typically obtained through both conventional methods—such as steam distillation, hydro-distillation, solvent extraction, and hydro-diffusion—and advanced non-conventional techniques, including solvent-free microwave extraction, supercritical fluid extraction, and subcritical liquid extraction [7,14]. The essential oils of Moroccan thyme species are notably rich in monoterpenes such as thymol, carvacrol, *p*-cymene, and γ-terpinene [15,16]. These oils exhibit significant variation in their phytochemical profiles, even within the same species, a phenomenon often attributed to ecogeographic factors. Climatic conditions and soil composition play a crucial role in shaping their chemical diversity [17,18]. Additionally, these essential oils are recognized for their diverse biological activities, contributing to their widespread use in traditional medicine and other applications.

The therapeutic potential of *Thymus* species has been extensively studied, particularly Moroccan thyme, which boasts rich biodiversity and unique biological properties. Many endemic species demonstrate strong antimicrobial, antioxidant, and antifungal activities in both in vitro and in vivo studies [16,19,20,21]. For example, *T. broussonetii* and *T. maroccanus* are rich in thymol and carvacrol, which give these species strong antioxidant activity, essential for neutralizing free radicals and protecting against oxidative stress [22]. These essential oils also demonstrated antitumor activity [23]. Furthermore, these thyme species exhibit significant enzyme-inhibitory activities against acetylcholinesterase and tyrosinase [24,25], highlighting their potential therapeutic value for neurodegenerative diseases and pigmentation disorders. Similarly, studies have reported that *T. zygis* subsp.* gracilis* essential oil possesses antioxidant, anti-proliferative, anti-inflammatory, and antihyperglycemic effects [26,27], as well as antimicrobial activity [21]. Moreover, *T. satureioides* essential oil shows promise across a broad spectrum of activities, including antibacterial, anti-inflammatory, antioxidant, antitumor, antifungal, antiviral, and antiparasitic properties [14,28,29].

Besides their therapeutic properties, plant-derived compounds have demonstrated potential in insecticidal activities [15,30]. Several studies have reported the contact toxicity of EOs from the Lamiaceae family against aphids. Essential oils from *Thymus capitatus*, *Thymus vulgaris* L., *Rosmarinus officinalis*, and *Eucalyptus globulus* have shown notable efficacy in reducing aphid populations when applied in different formulations [31,32,33]. Additionally, EOs from *Mentha piperita* L., *Mentha pulegium* L., and *Ocimum basilicum* L. demonstrated high efficacy in contact tests against *Lipaphis pseudobrassicae* Davis, *Myzus persicae*, and *A. gossypii* [34]. Oils from *Mentha longifolia* L. also exhibited contact activity against *Aphis craccivora* Koch, impacting its development, survival, and reproduction [35]. Despite Morocco’s diversity of approximately 14 thyme species [36], however, the aphicidal properties of Moroccan thyme remain unexplored.

The objectives of this study were multifaceted. First, we aimed to analyze the variation in chemical compounds among five Moroccan thyme varieties collected from four distinct locations. We also evaluated the EOs’ aphicidal activity against *A. gossypii*, as well as their antioxidant properties and inhibitory effects on acetylcholinesterase, tyrosinase, and α-glucosidase enzymes.

## 2. Results

The mineralogical analysis of thyme species is crucial due to the essential role minerals play in biological functions, such as tissue formation, metabolic processes, and maintaining physiological balance. Key minerals like calcium (Ca), phosphorus (P), magnesium (Mg), sulfur (S), potassium (K), and sodium (Na) contribute significantly to these processes, while others like copper (Cu) and zinc (Zn) offer additional health benefits, such as reducing cardiovascular risk factors. For instance, potassium regulates blood pressure, magnesium aids in muscle relaxation and prevents arrhythmias, and calcium and zinc support bone structure and antioxidant defenses. The mineral richness of thyme, beyond its characteristic chemical compounds, enhances its nutritional and medicinal value, highlighting the importance of comprehensive mineralogical analysis for its evaluation and valorization in human health contexts [37,38,39,40,41]. This study focuses on evaluating the mineral composition of five thyme species from distinct Moroccan regions, exploring the impact of geographical and climatic variations. The findings aim to deepen our understanding of Moroccan thyme’s nutritional and therapeutic properties, enriching its value in traditional and modern applications.

The mineral analysis of the aerial parts of five thyme species (Table 1) demonstrated important variations in mineral content based on different harvesting locations. *T. willdenowii* Boiss exhibited the highest concentration of essential minerals, notably calcium (712.2 mg/kg), followed by potassium (420.7 mg/kg), magnesium (150.7 mg/kg), phosphorus (30.3 mg/kg), and moderate iron (9.1 mg/kg). In comparison, *T. satureioides* showed substantial levels of calcium (533.8 mg/kg), potassium (360.4 mg/kg), and phosphorus (38.0 mg/kg), along with a higher iron content (14.0 mg/kg), indicating a balanced mineral profile. Similarly, *T. broussonetii* subsp. *broussonetii* contained important amounts of calcium (518.2 mg/kg), potassium (255.5 mg/kg), and iron (15.2 mg/kg). In contrast, *T. maroccanus* had lower calcium (450.6 mg/kg) and potassium (340.7 mg/kg) levels, with magnesium (111.6 mg/kg) and trace iron (3.3 mg/kg), suitable for moderate mineral intake. Lastly, *T. zygis* revealed intermediate calcium (480.3 mg/kg) and potassium (403.5 mg/kg) levels, along with magnesium (123.1 mg/kg), although it showed slightly less diversity in mineral composition compared to other species. Collectively, these thyme species present distinct yet complementary mineral profiles.

### 2.1. Yield and Analysis of Essential Oils

The oil yields of the studied species exhibit important variations, underscoring their potential for EO extraction. Among these, *T. willdenowii*, harvested from Lahri in Khenifra, stands out as the most prolific, boasting an impressive oil yield of 3.7%, indicating its rich EO content. Close behind, *T. zygis*, collected from Ait Ishak in Khenifra, shows a substantial oil yield of 2.1%. In contrast, *T*. *maroccanus*, sourced from Ait Ourir in Marrakech, and *T. satureioides*, found in Ijoukak, Marrakech, yield moderate amounts of 1.8% and 1.6%, respectively. Meanwhile, *T. broussonnetii* subsp. *broussonnetii*, collected from Ounagha in Essaouira, produces the lowest yield at 1.4% (Table 2). These findings suggest that *T. willdenowii* and *T. zygis* may be more advantageous for commercial oil extraction applications.

The GC-MS analysis of the *Thymus* species EOs enabled the identification of multiple compounds, representing 92.9% to 98.8% of the total chemical composition (Table 2). The oils were primarily dominated by oxygenated monoterpenes (40.1–58.9%) and monoterpene hydrocarbons (30.7–46.6%), with smaller proportions of hydrocarbon sesquiterpenes (2.2–8.1%) and oxygenated sesquiterpenes (0.5–4.3%). Each species exhibited distinct profiles with specific primary compounds. *T. willdenowii* was rich in thymol (48.3%), γ-terpinene (15.9%), and *p*-cymene (13.2%), with minor constituents like linalool (3.4%) and borneol (3.2%). In *T. zygis* subsp. *gracilis*, thymol (41.5%) was the major compound, followed by *p*-cymene (23.0%) and γ-terpinene (8.9%), along with borneol (4.8%) and linalool (3.7%). *T. broussonnetii* subsp. *broussonnetii* showed a balanced composition, with thymol (17.4%), borneol (16.8%), and *p*-cymene (16.9%), along with camphene (7.2%) and α-pinene (9.1%). *T. maroccanus* contained high concentrations of thymol (38.1%), *p*-cymene (18.8%), and carvacrol (12.9%), with minor levels of α-terpineol (3.5%) and linalool (1.0%). Finally, *T. satureioides* was notable for its high borneol content (19.4%) and moderate levels of thymol (13.8%), camphene (12.5%), and *p*-cymene (10.4%), with α-pinene (6.8%) and *trans*-caryophyllene (6.1%) also present (Figure 1). This unique chemical diversity across species underscores their potential for applications in flavoring, antimicrobial, and therapeutic formulations.

### 2.2. Antioxidant Properties

The evaluation of antioxidant activity for EO derived from *T. willdenowii*, *T. zygis* subsp. *gracilis*, *T. broussonnetii* subsp. *broussonnetii*, *T. maroccanus*, and *T. satureioides* using DPPH, FRAP, and *β*-carotene methods reveals important variations in efficacy compared to the standards, gallic acid and BHT (Table 3).

In the DPPH assay, *T. willdenowii* demonstrated the strongest antioxidant activity with an IC_50_ value of 51.3 µg/mL, outperforming gallic acid (47.7 µg/mL) and significantly exceeding BHT (169.0 µg/mL). The EOs from *T. zygis* subsp. *gracili*s exhibited moderate antioxidant activity with an IC_50_ of 70.6 µg/mL, while *T. broussonnetii* subsp. *broussonnetii* showed a further decrease in radical scavenging potential with an IC_50_ of 92.1 µg/mL. Conversely, *T. maroccanus* (114.8 µg/mL) and *T*. *satureioides* (146.7 µg/mL) exhibited weaker DPPH scavenging abilities, indicating a reduced capacity to neutralize free radicals compared to both gallic acid and BHT. Overall, *T. willdenowii* stood out as the most effective EO in this assay, highlighting its potential as a natural antioxidant source.

The FRAP assay results further support the superior antioxidant properties of *T. willdenowii*, with an IC_50_ of 34.6 µg/mL, demonstrating robust reducing power. This value is notably lower than that of gallic acid (16.3 µg/mL), suggesting that while *Thymus willdenowii* is effective, it is slightly less potent in terms of reducing capacity compared to the standard. *T. zygis* subsp. *gracilis* showed moderate reducing power (47.4 µg/mL), while *T. broussonnetii* subsp. *broussonnetii* (72.1 µg/mL), *T. maroccanus* (114.3 µg/mL), and *T. satureioides* (145.4 µg/mL) exhibited diminishing efficacy. The increasing IC_50_ values in these EOs reflect a decreasing ability to reduce ferric ions, indicating that *T. willdenowii* remains the most promising EO, while the other oils may require further optimization to enhance their antioxidant capabilities.

In the *β*-carotene assay, *T*. *willdenowii* again exhibited the highest antioxidant activity with an IC_50_ of 33.9 µg/mL, significantly lower than gallic acid (72.2 µg/mL), indicating its strong ability to inhibit oxidation. *T. zygis* subsp. *gracilis* (52.0 µg/mL) and *T. broussonnetii* subsp. *broussonnetii* (71.4 µg/mL) also showed moderate inhibition of *β*-carotene oxidation, while *T. satureioides* (98.4 µg/mL) and *T. maroccanus* (127.5 µg/mL) revealed considerably weaker antioxidant effects. The varying IC_50_ values among these EOs underscore *T. willdenowii’*s exceptional capacity to protect *β*-carotene from oxidation, further supporting its potential as a valuable natural antioxidant. Overall, *T. willdenowii* consistently outperformed both the standards and the other EOs across all three methods, indicating its promising application in food preservation and health-related products.

### 2.3. Aphicidal Effect of EOs

The contact toxicity of different EOs against adult *A*. *gossypii* is shown in Table 4, where control treatments demonstrated no mortality. The toxicity levels of the tested EOs varied significantly across species. Among the five EOs tested, *T. willdenowii* exhibited the highest efficacy, with an LC_50_ value of 6.2 µL/mL, indicating the strongest toxic effect on *A. gossypii* adults. *T. maroccanus* and *T. zygis* showed moderate toxicity, with LC_50_ values of 7.2 and 7.1 µL/mL, respectively. On the other hand, the EOs from *T. broussonnetii* and *T. satureioides* demonstrated the lowest toxicity, with LC_50_ values of 10.9 and 15.9 µL/mL, respectively, indicating a less effective impact on aphid mortality. This variation highlights the potential of specific EOs as targeted pest control agents based on their differential toxicity levels.

### 2.4. Enzyme Inhibitory Activities

The inhibitory effects of EOs extracted from five thyme species, sourced from various locations, were evaluated on acetylcholinesterase (AChE), tyrosinase, and α-glucosidase enzymes. This in vitro analysis quantified inhibition percentages using the Ellman method [43], with detailed findings outlined in Table 5. Generally, a higher concentration of EOs corresponded with an increase in inhibitory activity. According to the classification by Custódio et al. [44], enzyme inhibition levels are defined as potent (>50%), moderate (30–50%), low (<30%), or negligible (<5%).

The studied EOs demonstrated concentration-dependent inhibitory effects on AChE (Table 5), showing a marked increase in inhibition with higher doses. At 1 mg/mL, *T. willdenowii* EO exhibited the highest AChE inhibition, reaching 96.4%, significantly surpassing the positive control (74.7%). *T. zygis* subsp. *gracilis* and *T. broussonnetii* subsp.* broussonnetii* also showed strong inhibition, with rates of 90.6% and 90.5%, respectively. Although *T. satureioides* and *T. maroccanus* exhibited slightly lower inhibition at this concentration (83.4% and 83.1%), their effects remained notable. The exceptional inhibition observed with *T. willdenowii* suggests that specific active compounds or synergistic interactions may contribute to its potency.

The tyrosinase inhibition activity of these EOs also follows a dose-dependent pattern. At 1 mg/mL, *T. willdenowii* achieved an inhibition rate of 89.6%, marginally higher than the positive control’s 87.6%. Essential oils of *T. zygis* subsp. *gracilis* and *T. satureioides* exhibited similar effectiveness at this concentration, with inhibition rates of 86.9% and 86.4%, respectively. Additionally, *T. maroccanus* and *T. broussonnetii* subsp. *broussonnetii* showed robust activity with inhibition rates of 85.2% and 77.5%, respectively, at 1 mg/mL.

Regarding α-glucosidase inhibition, *T. willdenowii* EO once again showed the greatest efficacy at 1 mg/mL, with an inhibition rate of 89.5%, closely approaching the positive control’s 89.9%. *Thymus zygis* subsp. *gracilis* followed with 86.4% inhibition, while *T. broussonnetii* subsp. *broussonnetii* also performed well, achieving 79.4%. Effective inhibition was also observed in *T. maroccanus* and *T. satureioides*, with rates of 82.0% and 84.1%, respectively. These results underscore the promising α-glucosidase inhibitory properties of *T. willdenowii* and *T. zygis* subsp. *gracilis*, which may be beneficial for managing carbohydrate metabolism and glycemic control.

## 3. Discussion

The mineral composition analysis of the five thyme species reveals important interspecies and locational variation in essential mineral content, all within the permissible limits established by the World Health Organization [45], affirming their safety and potential as dietary mineral sources. Notably, *Thymus willdenowii* Boiss exhibited the highest levels of calcium, potassium, and magnesium, with a balanced presence of phosphorus and iron, aligning with its role as a valuable source of these minerals for bone health and metabolic support [46]. Similarly, *T. satureioides* displayed high calcium and potassium concentrations, with a notably elevated iron content, suggesting potential applications in dietary interventions aimed at addressing iron deficiencies [46]. The substantial calcium and potassium levels in *T. broussonetii* further highlight its potential for nutraceutical use, while *T. maroccanus*, with moderate calcium and potassium and minimal iron, may suit individuals needing lower mineral intakes. *Thymus zygis* subsp. *gracilis*, with its intermediate mineral profile, could serve well in diets requiring balanced, moderate mineral content without exceeding WHO limits [46,47]. Collectively, these thyme species offer a safe, nutritionally diverse selection of minerals, underscoring their value as natural mineral sources for dietary supplementation and supporting sustainable health applications.

The EO yields observed in this study for *Thymus* species showed a marked variation, reflecting differences potentially due to species-specific traits and environmental influences tied to geographic origins. *Thymus willdenowii*, harvested from Lahri in Khenifra, exhibited the highest yield at 3.7%, exceeding the typical range of 0.3–3.0% reported in other studies [46,48,49]. Similarly, *T. zygis* subsp. *gracilis* from Ait Ishak in the Khenifra region showed a yield of 2.1%, placing it within the upper range of results reported by Benabderrahmane et al. [50], who found a yield of 1.6%. In contrast, *T. maroccanus* collected from Ait Ourir in Marrakech showed a moderate yield of 1.8%, which falls within the range of 1.0–3.5% documented by Jamali et al. [51] for various ecotypes in similar environments. *Thymus satureioides* from Ijoukak in Marrakech yielded 1.6%, which remains relatively low compared to the range of 2.3 to 2.4% reported by El-Bakkal et al. [52]. Similarly, *T. broussonnetii,* harvested in Ounagha, Essaouira, showed the lowest yield at 1.4%, significantly lower than the 3.3% reported by Tagnaout et al. [53]. These results emphasize the important role of geographic origin in determining EO productivity within the same species, highlighting the impact of local environmental conditions on EOs yield optimization.

The EO of *T. willdenowii* revealed a chemotype predominantly composed of thymol (48.3%), γ-terpinene (15.9%), and *p*-cymene (13.2%), with minor components such as linalool (3.4%) and borneol (3.2%). These results differ from the findings of Zeghib et al. [54], who identified 1,8-cineole (34.6%), camphor (18.5%), α-pinene (9.4%), and camphene (5.3%) as the major constituents, suggesting the presence of distinct chemotypes within *T. willdenowii*. Differences in EO composition may be affected by factors like genetic variation, environmental conditions, and seasonal or geographical variations [55]. For *T. zygis* subsp. *gracilis*, thymol (41.5%) was the primary component, followed by *p*-cymene (23.0%) and γ-terpinene (8.9%), with minor compounds like borneol (4.8%) and linalool (3.7%). This profile contrasts with the results of Radi et al. [48], who found a significantly different composition, with carvacrol (52.5%), o-cymene (23.1%), and thymol (9.7%) as the dominant components. In our analysis of *T*. *broussonnetii* subsp. *broussonnetii*, we observed a balanced EO composition with thymol (17.4%), borneol (16.8%), and *p*-cymene (16.9%) as the major constituents, along with camphene (7.2%) and α-pinene (9.1%). This differs from the profile identified by Tagnaout et al. [53], which was dominated by carvacrol (60.8%), with lesser amounts of thymol (12.9%), *p*-cymene (6.2%), and γ-terpinene (4.5%) in *T*. *broussonnetii*. For *T*. *maroccanus*, we identified high levels of thymol (38.1%), *p*-cymene (18.8%), and carvacrol (12.9%), with minor quantities of α-terpineol (3.5%) and linalool (1.0%). In contrast, El Bouzidi et al. [56] reported a profile where carvacrol, borneol, γ-terpinene, and *p*-cymene were the primary components. For *T. satureioides*, our analysis showed an important concentration of borneol (19.4%), along with moderate levels of thymol (13.8%), camphene (12.5%), *p*-cymene (10.4%), α-pinene (6.8%), and *trans*-caryophyllene (6.1%), aligning closely with the results of several studies [57,58], who similarly reported a predominance of borneol (26.4%) and thymol (11.2%). Differences in essential oil composition between studies arise from genetic variation, environmental factors (altitude, soil, climate), geographical and seasonal influences, harvesting practices, and extraction methods. These variables underscore the dynamic nature of EO composition and the need to account for them in comparative analyses [54,57,58].

The antioxidant potential of various *Thymus* species was comprehensively assessed using three methods: DPPH, FRAP, and the *β*-carotene bleaching test, affirming the promising role of these EOs as natural antioxidants. *Thymus willdenowii* stood out with remarkable DPPH radical scavenging activity, achieving an IC_50_ of 51.3 µg/mL, an effect attributed to its high thymol content, known for potent antioxidant properties, or possibly due to synergistic interactions among its compounds [15]. Previous studies consistently show that oils rich in thymol exhibit enhanced antioxidant activity, underscoring the importance of chemical composition in antioxidant efficacy [59,60,61,62].

In the FRAP assay, *Thymus willdenowii* again demonstrated strong reducing power, with an IC_50_ of 34.6 µg/mL, indicating substantial ferric ion reduction capacity, though slightly less potent than gallic acid (16.3 µg/mL). This aligns with evidence correlating phenolic content to reducing power [60,61,62]. The *β*-carotene bleaching test further highlighted *T. willdenowii*’s superior antioxidant activity, with an IC_50_ of 33.9 µg/mL significantly lower than that of gallic acid (72.2 µg/mL)—showcasing its strong inhibition of oxidation. *Thymus zygis* subsp. *gracilis* (52.0 µg/mL) and *T*. *broussonnetii* (71.4 µg/mL) also showed noteworthy antioxidant performance, while *T*. *satureioides* (98.4 µg/mL) and *T*. *maroccanus* (127.5 µg/mL) exhibited lower antioxidant activity, likely due to lower concentrations of active compounds [25,62,63]. These findings underscore *T. willdenowii* as an especially potent source of natural antioxidants, reinforcing the critical impact of chemical composition on antioxidant effectiveness across *Thymus* species.

Our findings revealed that the lethal concentrations of the EOs varied among the species tested, with *Thymus willdenowii* EO showing the highest contact toxicity against *A. gossypii* adults 24 h post-treatment, as indicated by an LC_50_ value of 6.2 µL/mL. This was followed by *T. maroccanus* and *T. zygis*, which exhibited moderate toxicity, with LC_50_ values of 7.2 µL/mL and 7.1 µL/mL, respectively. This toxicity was likely due to the EO’s high concentration of monoterpenes, particularly thymol, γ-terpinene, and *p*-cymene, which have demonstrated strong insecticidal effects across multiple insect species [30,64]. Similarly, Alahyane et al. [65] found that *Thymus munbyanus*, *Thymus willdenowii*, and *Lavandula maroccana* are rich in EOs containing thymol, as well as *p*-cymene EOs, which exhibit high toxicity against Varroa mites. However, research by Benchouikh et al. [66] and Yakhlef et al. [67] suggests that the toxic impact of EOs on insects is not solely due to the main compounds, as minor compounds may contribute synergistically to enhance the overall effect. In insects such as aphids, EO toxicity appears to involve a combination of lethal and sublethal actions. Specifically, the chemical components of EOs disrupt the central nervous system of insects, particularly by interfering with GABA-gated chloride ion channels and octopamine pathways, which induce neurotoxic effects such as paralysis and death [68,69]. Moreover, EOs act as inhibitors of acetylcholinesterase activity [16,70], which is an essential neurotransmitter in both the central and peripheral nervous systems of insects. Beyond neurotoxicity, EOs also affect the growth and reproductive processes of many species belonging to the Aphididae family. For example, the EO of *Tagetes minuta* L. significantly reduced the fecundity of *Acyrthosiphon pisum* Harris, *Myzus persicae* (Sulzer), and *Aulacorthum solani* Kaltenbach [71]. Additionally, EOs from *Origanum majorana* L., *Mentha pulegium* L., and *Melissa officinalis* L. have been shown to significantly reduce both the longevity and fecundity of *M*. *persicae* [72].

From an integrated pest management perspective, evaluating the effects of EOs on pest control also requires examining their impact on natural predators. The use of chemical controls often increases pest resistance and adversely affects natural predators, diminishing their capacity to help manage pest populations and potentially leading to pest outbreaks. However, studies have shown that several EOs are less harmful to predators than to target pests. For instance, Sayed et al. [73] investigated the insecticidal activities of *Mentha piperita*, *Mentha longifolia*, *Salvia officinalis*, and *Salvia rosmarinus* EOs on *Aphis punicae* and its predator, *Coccinella undecimpunctata*, finding that the EOs were highly toxic to the aphid, with LC_50_ values four times higher than for the coccinellid. Similarly, Papadimitriou et al. [74] found that the EO of *Mentha pulegium* was highly lethal to *A. gossypii* at a concentration of 1000 µL/L, yet had no toxic effects on its predator, the *Nesidiocoris tenuis*. Ebadollahi et al. [75] found that the EO of *Satureja intermedia* showed important contact toxicity against *Aphis nerii* Kaltenbach, yet was safe for its predator, *Coccinella septempunctata* L. This suggests that natural insect predators may have higher tolerance to various plant EOs than the target pest species.

The EOs from various *Thymus* species demonstrated important concentration-dependent inhibitory effects on key enzymes such as acetylcholinesterase (AChE), tyrosinase, and α-glucosidase, highlighting their potential therapeutic applications. *Thymus willdenowii* exhibited the highest AChE inhibition at 96.4% at a concentration of 1 mg/mL, likely due to its high thymol content (48.3%), which is known to interact effectively with the enzyme’s active site, enhancing cognitive function and neuroprotection [76,77,78,79]. In terms of tyrosinase inhibition, *T*. *willdenowii* achieved a leading inhibition rate of 89.6%. This effect is attributed to its potent EOs components, such as thymol, γ-terpinene, and *p*-cymene, which effectively inhibit melanin synthesis, underscoring its potential in cosmetic applications [80,81,82]. Furthermore, regarding α-glucosidase inhibition, *T*. *willdenowii* exhibited a remarkable efficacy of 89.5%, closely followed by *T. zygis* subsp. *gracilis* at 86.4%, suggesting that their high levels of thymol (48.3% and 41.5%) can significantly modulate carbohydrate metabolism [83,84]. The strong enzyme inhibitory activities observed in these *Thymus* species can be attributed to their rich EO composition, particularly high levels of thymol, γ-terpinene, *p*-cymene, and carvacrol. This highlights their promising potential as natural sources of bioactive compounds.

## 4. Materials and Methods

### 4.1. Plant Material and Essential Oil Isolation

The aerial parts of five *Thymus* species (T1, T2, T3, T4, and T5) were collected during the full flowering period (June–July 2021) from five distinct ecological regions in Morocco. The taxonomic identification of the plant material (Table 6) was performed based on the practical flora of Morocco [85] and verified at the herbarium of the Faculty of Science and Technology in Errachidia. After collection, the plants were air-dried at room temperature. Essential oils were extracted separately from 100 g of dried plant material for each sample using hydrodistillation in a Clevenger-type apparatus (purchased in 2021) for 3 h [86]. The resulting EOs were dried over anhydrous sodium sulfate, filtered, and stored at −4 °C until further analysis.

### 4.2. Plant Mineral Analysis

The aerial parts of the studied plants (T1, T2, T3, T4, and T5) were individually processed through a series of precise analytical steps. Initially, each sample was thoroughly washed with distilled water and then oven-dried at 80 °C until a stable weight was reached. The dried plant material was finely ground using a mortar, and 0.5 g of the resulting powder was subjected to mineralization. This process involved a digestion mixture of 2 mL H_2_SO_4_ (98%), 6 mL HNO_3_ (65%), and 6 mL H_2_O_2_ (35%), which was heated in a sand bath for 30 min. After digestion, the suspension was filtered, and the filtrate volume was adjusted to 25 mL using 0.1 M nitric acid [46]. The metal content in the prepared solution was analyzed using a plasma emission spectrometer, namely, the JOBIN-YVON 70 ICP (Inductively Coupled Plasma) ULTIMA and JY 70 models, developed by Horiba Scientific (formerly Jobin-Yvon, France, 16–18 rue du Canal, 91165 Longjumeau Cedex).

### 4.3. GC Analysis

The analysis of studied essentials oils was performed using a Perkin-Elmer Autosystem XL gas chromatograph (PerkinElmer, Inc., 940 Winter Street, Waltham, MA 02451, USA) equipped with dual flame ionization detectors (FID) (PerkinElmer, Inc., 940 Winter Street, Waltham, MA 02451, USA) and two fused-silica capillary columns: Rtx-1 (polydimethylsiloxane) and Rtx-Wax (polyethyleneglycol), each measuring 60 m × 0.22 mm (I.D.) with a film thickness of 0.25 µm. The oven temperature was programmed to increase from 60 °C to 230 °C at a rate of 2 °C/min, followed by an isothermal hold at 230 °C for 35 min. Both the injector and detector temperatures were maintained at 280 °C. Samples were injected in split mode (1:50) with helium as the carrier gas at a flow rate of 1 mL/min, and an injection volume of 0.2 µL of pure oil was used. Retention indices (RI) were calculated relative to a series of n-alkanes (C5–C30) using the Van den Dool and Kratz equation [87], with the assistance of Perkin-Elmer software (version 6.3.2). Additionally, retention indices on the Joulain scale (RIj) were evaluated to confirm the identification of compounds in the studied essential oils [42]. The relative concentrations of the components were determined from the GC peak areas without applying correction factors.

### 4.4. GC-MS Analysis

For GC-MS analysis, the samples were also analyzed using a Perkin-Elmer Turbo mass detector (quadrupole) coupled to a Perkin-Elmer 88 Autosystem XL GC (PerkinElmer, Inc., 940 Winter Street, Waltham, MA 02451, USA), fitted with the same fused-silica capillary columns (Rtx-1 and Rtx-Wax). Helium was used as the carrier gas at a flow rate of 1 mL/min. The ion source temperature was set at 150 °C, and the oven temperature was programmed identically to the GC-FID analysis (60 °C to 230 °C at 2 °C/min, followed by an isothermal hold at 230 °C for 35 min). The injector temperature was maintained at 280 °C, and electron ionization (EI) was performed at 70 eV, with mass spectra recorded over a mass range of 35–350 atomic mass units (amu). Samples were injected in split mode (1:80) with a volume of 0.2 µL of pure oil.

### 4.5. Insect Rearing

A colony of cotton aphid (*Aphis gossypii* Glover) was obtained from the Agafay orchards’ Insectarium in Marrakech, Morocco (N 31°30′04.0″, W 8°14′54.4″). The insect colony had been reared for multiple generations under laboratory conditions, following the protocol outlined by Alahyane et al. [88]. The aphids were continuously maintained on *Phaseolus vulgaris* L. under controlled conditions of 23 ± 2 °C; 60 ± 5% relative humidity (RH); and a 14:10 h (light: dark) photoperiod, illuminated by white LED lamps.

### 4.6. Antioxidant Activities

#### 4.6.1. DPPH Assay

The antioxidant activity of studied plant EOs was assessed using the DPPH (2,2-diphenyl-1-picrylhydrazyl) free radical scavenging assay, as outlined by Ouknin et al. [89]. Briefly, various dilutions of the EOs and their key active components were mixed with a 0.4 mM methanolic solution of DPPH in a 50 μL to 5 mL ratio. After 30 min of incubation in darkness, the absorbance of the mixtures was measured at 517 nm using a Uviling 9400 (SECOMAM) spectrophotometer. Butylated hydroxytoluene (BHT) and gallic acid were used as positive controls. The radical scavenging activity was calculated using the following Formula (1):(1)$$DPPH\;Scavenging\;effect\left(\%\right)=\left(\frac{A_{0}-A_{1}}{A_{0}}\right)\times 100$$

A_0_ represents the control absorbance measured at 30 min, while A_1_ corresponds to the absorbance of the sample recorded after 30 min. All experiments were conducted in triplicate.

#### 4.6.2. Reducing Power Determination (FRAP)

The reductive capacity was assessed by monitoring the conversion of Fe^3+^ to Fe^2+^ in the presence of the studied EOs, following the protocol outlined by Oyaizu [90]. The EOs and reference substances were diluted in ethanol, creating concentration ranges from 20 µg/mL to 400 µg/mL for EOs and from 5 µg/mL to 100 µg/mL for the control substances. Each concentration was combined with phosphate buffer (2.5 mL, 0.2 M, pH 6.6) and potassium ferricyanide [K_3_Fe(CN)_6_] (2.5 mL, 1%). The resulting mixtures were incubated at 50 °C for 20 min. Trichloroacetic acid (2.5 mL, 10%) was then added, and the mixtures were centrifuged for 10 min at 3000 rpm. The upper layer (2.5 mL) was mixed with distilled water (2.5 mL) and FeCl_3_ (0.5 mL, 0.1%), and absorbance was measured at 700 nm using a spectrophotometer.

The concentration of EOs required to achieve 0.5 absorbance (IC_50_) was determined by plotting absorbance values against the corresponding oil concentrations. BHT and gallic acid were used as standard references. The experiment was performed in triplicate, and IC_50_ values were expressed as means ± SD.

#### 4.6.3. β-Carotene Bleaching Test

The antioxidant potential of the EOs under study was evaluated by their capacity to prevent *β*-carotene bleaching in a linoleic acid system, using a method modified from Ouknin et al. [89]. Gallic acid and BHT were employed as positive controls, while ethanol was used as the negative control, substituting for the EOs. The antioxidant effectiveness of the five thyme species was quantified by measuring the degree of *β*-carotene bleaching, calculated using Equation (2).
(2)$$I\%=\left(\frac{A_{\beta -carotene\;after\;2h}}{A_{initial\beta -carotene}}\right)\times 100$$
where A*_β_*_-carotene after 2h_ represents the absorbance of *β*-carotene remaining in the samples after a 2 h assay, and A_initial *β*-carotene_ refers to the absorbance of *β*-carotene at the start of the experiment. All assays were performed in triplicate. The oil concentration required to achieve 50% inhibition (IC_50_) was determined by plotting the percentage of inhibition against the corresponding oil concentration.

### 4.7. Aphicidal Activity of EOs

To evaluate the aphicidal activity of the EO, contact toxicity bioassays were performed. For this, a haricot common leaf was placed in a 70 mm diameter Petri dish with a pair of moistened filter paper disks at the bottom. Then, 10 apterous adults of *A. gossypii* were placed on the leaf and left to settle for 10 min, and then the EO to be tested was sprayed over the aphids. For the bioassays, all the EO were prepared at three doses: 4, 10, and 15 μL/mL in ethanol. A total volume of 1 mL was prepared for each EO and completely sprayed over the aphids using a hand-held micro-applicator. Similar treatments with only ethanol were used as controls. Once treatments were completed, the Petri dishes were placed into a growth chamber at 23 ± 2 °C and 60 ± 5% RH. Contact toxicity bioassays were replicated four times. Mortality was recorded 24 h after treatment; aphids were considered dead if they showed no response when gently touched with a soft fine brush.

### 4.8. Enzyme Inhibitory Properties

#### 4.8.1. AChE Inhibition

Cholinesterase inhibitory activity was evaluated using a microplate reader, following a modified approach based on the method by Ouknin et al. [16]. The assay procedure involved mixing 25 μL of 15 mM ATCI with 125 μL of 3 mM DTNB, then adding 50 μL of 100 mM phosphate buffer (pH 8.0); 25 μL of EO at concentrations of 0.25, 0.5, 0.75, and 1 mg/mL; along with either 25 μg/mL of galanthamine (positive control) or buffer alone (negative control). AChE (0.28 U/mL) was then added, and absorbance was recorded at 405 nm over 5 min. To determine the percentage of AChE inhibition, a buffer-only control was included, and all tests were conducted in triplicate.

#### 4.8.2. Tyrosinase Inhibition

Tyrosinase inhibitory activity was assessed via spectrophotometry, following the procedure outlined by Masuda et al. [91]. The assay involved adding EOs at concentrations of 0.25, 0.5, 0.75, and 1 mg/mL, or phosphate buffer as a blank, to 80 μL of phosphate buffer (pH 6.8). Next, 40 μL of L-DOPA and 40 μL of tyrosinase enzyme were introduced. Kojic acid, at a concentration of 200 μg/mL, served as the reference standard. Absorbance was recorded at 475 nm, and the inhibitory activity percentage was determined. Each experiment was conducted in triplicate.

#### 4.8.3. α-Glucosidase Inhibition

The evaluation of α-glucosidase inhibitory activity was performed following the procedure described by Kwon et al. [92]. Different concentrations of the EOs (0.25, 0.5, 0.75, and 1 mg/mL) or acarbose (1 mg/mL, used as a positive control) were combined with 50 μL of phosphate buffer (0.1 M, pH 6.9) containing yeast α-glucosidase (1.0 U/mL). The mixture was incubated at room temperature for 10 min before adding 50 μL of a *p*-nitrophenyl-α-D-glucopyranoside solution (5 mM). After a further 10 min incubation, absorbance was recorded at 405 nm to calculate the percentage inhibition of α-glucosidase. Each assay was performed in triplicate.

### 4.9. Statistical Analysis

Statistical analyses were performed using SPSS 25.0 software, which facilitated descriptive data analysis, including the calculation of means and standard deviations (SDs). The results of the enzyme inhibition effects of the essential oils were analyzed using univariate analysis of variance (ANOVA), followed by Tukey’s test (*p* ≤ 0.05) to identify important differences among the various concentrations and individual enzymes. Lethal concentrations (LC_50_ values) were calculated using probit analysis [93], with chi-squared values and 95% confidence intervals also calculated.

## 5. Conclusions

This study highlights the mineral composition, oil yield, and bioactivity of EOs from five Moroccan thyme species: *T. willdenowii*, *T. satureioides*, *T. broussonetii*, *T. maroccanus*, and *T. zygis* subsp. *gracilis*. The mineral analysis revealed considerable variation across species, with *T. willdenowii* containing notably high levels of essential minerals like calcium (712.2 mg/kg) and potassium (420.7 mg/kg), highlighting its nutritional potential. Oil yield results position *T. willdenowii* from Lahri, Khenifra, as the highest producer (3.7%), followed by *T. zygis* subsp. *gracilis* (2.1%). GC-MS analysis showed that these oils are rich in oxygenated monoterpenes and monoterpene hydrocarbons, including thymol, γ-terpinene, and *p*-cymene, with *T. willdenowii* showing the highest thymol content (48.3%) and *T. satureioides* notable for its high borneol (19.4%) concentration. Antioxidant testing highlighted strong bioactivity, with *T. willdenowii* achieving an IC_50_ of 51.3 µg/mL in the DPPH assay, surpassing standard antioxidants, indicating promise for food preservation and health applications. The EOs of *T. willdenowii*, *T. maroccanus*, and *T. zygis* show promise as effective alternatives for managing *A. gossypii*, with *T. willdenowii* demonstrating the highest toxicity in contact applications. These EOs could serve as potent biopesticides within integrated pest management strategies for *A. gossypii*. Enzyme inhibition assays further highlight *T. willdenowii*’s strong activity against acetylcholinesterase (96.4%), as well as tyrosinase and α-glucosidase, suggesting therapeutic potential in the treatment of neurodegenerative diseases and type 2 diabetes. These results identify *T. willdenowii*, *T. maroccanus*, and *T. zygis* as promising candidates for applications in nutraceutical, pharmaceutical, and agricultural fields, encouraging further research on enhancing their stability and efficacy through advanced encapsulation techniques for sustainable health and food solutions.

## Figures and Tables

**Figure 1 plants-14-00116-f001:**
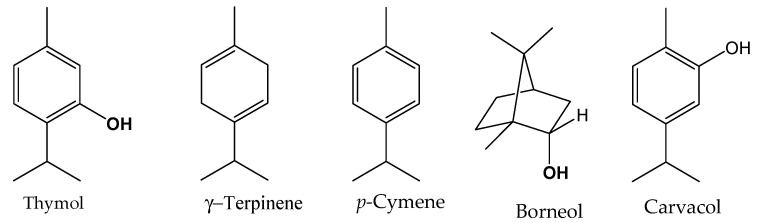
Main compounds identified in the studied essential oils.

**Table 1 plants-14-00116-t001:** Descriptive statistics of heavy metals and oligo-elements in *Thymus* species (mg/kg).

	*T. willdenowii* Boiss	*T. zygis*. subsp. *gracilis*	*T. broussonetii* subsp. *broussonetii*	*T. maroccanus*	*T. satureioides*
Al	3.4 ± 0.8 ^ab^	1.8 ± 0.1 ^a^	5.4 ± 1.3 ^b^	3.8 ± 0.5 ^ab^	4.4 ± 0.9 ^b^
B	0.3 ± 0.1 ^ab^	0.2 ± 0.1 ^a^	0.3 ± 0.1 ^bc^	0.2 ± 0.1 ^ab^	0.4 ± 0.1 ^c^
Ba	0.2 ± 0.1 ^ab^	0.4 ± 0.1 ^c^	0.1 ± 0.1 ^a^	0.3 ± 0.1 ^b^	0.2 ± 0.1 ^ab^
Ca	712.2 ± 16.0 ^d^	480.3 ± 10.0 ^ab^	518.2 ± 12.5 ^bc^	450.6 ± 21.0 ^ab^	534.0 ± 19.0 ^c^
Cr	0.7 ± 0.1 ^c^	0.1 ± 0.1 ^a^	0.5 ± 0.1 ^bc^	1.6 ± 0.1 ^d^	0.4 ± 0.1 ^b^
Fe	9.1 ± 4.4 ^ab^	1.9 ± 0.1 ^a^	15.2 ± 3.0 ^b^	3.3 ± 0.9 ^a^	14.0 ± 7.0 ^b^
K	420.7 ± 14.0 ^c^	403.5 ± 14.0 ^c^	255.5 ± 12.0 ^a^	340.7 ± 16.0 ^b^	360.4 ± 14.0 ^b^
Mg	150.7 ± 7.5 ^c^	123.1 ± 7.8 ^b^	100.4 ± 4.8 ^a^	111.6 ± 7.5 ^ab^	97.3 ± 5.7 ^a^
Mn	0.4 ± 0.1 ^ab^	0.3 ± 0.1 ^a^	0.8 ± 0.1 ^b^	0.4 ± 0.1 ^a^	0.9 ± 0.2 ^b^
Na	2.1 ± 0.1 ^a^	2.9 ± 0.6 ^ab^	3.1 ± 0.4 ^ab^	4.2 ± 1.1 ^bc^	4.7 ± 0.5 ^c^
P	30.3 ± 3.0 ^cd^	11.9 ± 0.5 ^a^	24.9 ± 1.5 ^bc^	19.2 ± 2.5 ^ab^	38.0 ± 5.2 ^d^
Si	3.9 ± 0.2 ^d^	2.3 ± 0.1 ^b^	1.2 ± 0.1 ^a^	2.8 ± 0.1 ^c^	4.5 ± 0.1 ^e^
Sn	0.1 ± 0.1 ^a^	0.1 ± 0.1 ^a^	0.1 ± 0.1 ^a^	0.1 ± 0.1 ^a^	0.1 ± 0.1 ^a^
Zn	0.5 ± 0.1 ^a^	0.4 ± 0.1 ^a^	1.1 ± 0.1 ^a^	3.8 ± 0.9 ^b^	0.7 ± 0.1 ^a^

Values are expressed as means ± SEs (n = 3). Identical small letters in the same line indicate no significant difference (*p* < 0.05).

**Table 2 plants-14-00116-t002:** GC-MS chemical profile of five thyme essential oils from different locations.

N ^a^	Components	^b^ RI j	^c^ RI a	^d^ RI p	T_1_ ^e^	T_2_ ^e^	T_3_ ^e^	T_4_ ^e^	T_5_ ^e^
1	Tricyclene	925	921	1016	–	–	–	–	0.3
2	*α*-Thujene	932	923	1026	0.6	0.2	1.1	–	0.1
3	*α*-Pinene	936	931	1026	0.8	1.2	9.1	8.6	6.8
4	Camphene	950	944	1072	1.0	1.6	7.2	0.3	12.5
5	Oct-1-en-3-ol	963	961	1448	0.3	0.4	–	0.8	–
6	Octan-3-one	964	966	1254	0.1	0.1	0.2	0.1	–
7	*β*-Pinene	978	971	1114	0.2	0.2	1.5	0.1	1.5
8	Myrcene	987	981	1163	1.6	1.2	2.4	0.1	1.5
9	*α*-Phellandrene	1002	1001	1168	0.2	0.2	0.2	0.2	0.2
10	*α*-Terpinene	1013	1011	1184	1.9	1.2	1.0	0.2	1.2
11	*p*-Cymene	1015	1013	1275	13.2	23.0	16.9	18.8	10.4
12	Limonene	1025	1020	1204	0.5	0.6	1.4	2.0	1.0
13	1,8 Cineole	1024	1020	1210	–	–	–	–	0.3
14	*β*-Phellandrene	1023	1020	1216	0.2	0.2	0.4	0.3	–
15	γ-Terpinene	1051	1049	1245	15.9	8.9	5.4	0.1	5.1
16	*trans*-Sabinene hydrate	1053	1055	1462	0.5	0.1	0.3	0.1	–
17	Nonen-3-ol	1058	1065	1522	–	0.1	–	–	–
18	*cis*-Linalool oxide THF	1072	1074	1441	0.1	0.2	0.1	0.1	–
19	Terpinolene	1082	1082	1286	0.1	0.2	0.2	–	0.3
20	Linalool	1086	1086	1547	3.4	3.7	0.9	1.0	2.3
21	Camphor	1123	1121	1506	–	–	0.3	0.5	0.3
22	Borneol	1150	1152	1700	3.2	4.8	16.8	0.5	19.4
23	Terpinen-4-ol	1164	1164	1600	0.6	0.9	0.3	0.6	–
24	*α*-Terpineol	1176	1174	1694	0.1	0.2	0.4	3.5	3.5
25	Carvacrol methylether	1226	1225	1603	0.1	0.1	–	–	1.2
26	Thymol	1267	1278	2180	48.3	41.5	17.4	38.1	13.8
27	Carvacrol	1278	1284	2207	3.2	3.2	3.4	12.9	0.3
28	*β*-Bourbonene	1292	1385	1519	–	–	–	–	0.2
29	*α*-Humulene	1312	1452	1666	–	–	–	–	0.3
30	γ-Muurolene	1325	1472	1687	–	–	–	–	0.2
31	*trans*-Caryophyllene	1421	1416	1600	1.9	2.1	0.4	0.3	6.1
32	Aromadendrene	1443	1440	1609	–	–	1.0	0.9	–
33	Ledene	1491	1494	1700	–	–	4.0	0.7	–
34	γ-Cadinene	1507	1508	1754	0.1	0.1	0.1	0.1	0.4
35	Calamenene	1517	1511	1830	0.1	0.1	0.1	0.2	–
36	δ-Cadinene	1520	1517	1754	0.1	0.1	0.2	0.1	0.5
37	Spathulenol	1572	1565	2120	0.1	0.2	0.6	1.2	–
38	Caryophyllene oxide	1578	1570	1980	0.4	1.1	0.2	0.5	0.5
Oxygenated monoterpenes					58.9	55.3	40.1	58.2	44.2
Hydrocarbon monoterpenes					37.0	38.8	46.6	30.7	41.5
Oxygenated sesquiterpenes					0.7	1.1	4.3	1.7	0.5
Hydrocarbon sesquiterpenes					2.2	2.4	2.5	2.3	8.1
Total identified (%)					98.8	97.6	93.5	92.9	95.3

^a^ Order of elution are provided on the apolar column (Rtx-1); ^b^ RI j = retention indices given by Joulain and König [42]; ^c^ RI a = retention indices on the apolar column (Rtx-1); ^d^ RI p = retention indices on the polar column (Rtx-Wax); ^e^ relative percentages of components (%) are calculated on GC peak areas on the apolar column (Rtx-1) except for components with identical RIa (concentration are given on the polar column); –: not detected. T1: *T. willdenowii*; T2: *T. zygis* subsp. *gracilis*; T3: *T. broussonnetii* subsp. *Broussonnetii*; T4: *T. maroccanus*; and T5: *T. satureioides.*

**Table 3 plants-14-00116-t003:** IC_50_ values (µg/mL) of the aerial parts of five essential oils of the genus *Thymus*.

Plants Code	DPPH (IC_50_ µg/mL)	FRAP (IC_50_ µg/mL)	*β*-Carotene (IC_50_ µg/mL)
T1	51.3 ± 0.5 ^a^	34.6 ± 0.6 ^b^	32.7 ± 1.2 ^a^
T2	70.6 ± 1.1 ^b^	47.4 ± 0.3 ^c^	52.0 ± 1.0 ^b^
T3	92.1 ± 1.1 ^c^	72.1 ± 0.8 ^d^	71.4 ± 1.3 ^c^
T4	114.8 ± 1.3 ^d^	114.3 ± 1.0 ^e^	127.5 ± 1.1 ^f^
T5	146.7 ± 0.7 ^e^	145.4 ± 1.9 ^f^	98.4 ± 0.6 ^e^
Gallic acid	47.7 ± 0.5 ^a^	16.3 ± 0.9 ^a^	72.2 ± 0.8 ^c^
BHT	169.0 ± 0.8 ^f^	36.9 ± 0.4 ^b^	92.6 ± 1.3 ^d^

Values are expressed as means ± SEs (n = 3). In the same column, values marked with different letters indicate significant differences (*p* < 0.05). T1: *T. willdenowii*; T2: *T. zygis* subsp. *gracilis*; T3: *T. broussonnetii* subsp. *broussonnetii*; T4: *T. maroccanus*; and T5: *T. satureioides.*

**Table 4 plants-14-00116-t004:** LC_50_ values (µL/mL) for the studied essential oils against adults of *A. gossypii*.

Essential Oils	LC_50_ (Confidence Interval Limits)	Intercept ± SE	Slope ± SE	χ^2^	*p*
*T. willdenowii* Boiss.	6.2 (5.1–7.5)	3.6 ± 0.4	1.8 ± 0.1	3.0	0.9
*T. zygi* subsp. *gracilis*	7.1 (5.4–9.2)	3.5 ± 0.6	1.8 ± 0.1	3.7	0.8
*T. broussonnetii* subsp. *broussonnetii*	10.9 (8.4–14.1)	3.1 ± 0.5	1.9 ± 0.1	10.3	0.7
*T. maroccanus*	7.2 (5.7–9.1)	3.2 ± 0.4	2.1 ± 0.1	0.4	1.0
*T. satureioides*	15.9 (12.9–19.7)	3.0 ± 0.4	1.6 ± 0.1	5.6	0.9

**Table 5 plants-14-00116-t005:** Enzymatic inhibition activity: acetylcholinesterase (AChE), tyrosinase, and α-glucosidase.

Plants Code	Essential Oil Doses (mg/mL)	AChE (%)	Tyrosinase (%)	α-Glucosidase (%)
T1	0.25	50.2 ± 0.6 ^c^	47.8 ± 0.3 ^b^	50.5 ± 0.4 ^b^
	0.5	62.3 ± 0.6 ^e^	62.2 ± 0.4 ^e^	62.4 ± 1.0 ^d^
	0.75	80.3 ± 0.9 ^kl^	75.5 ± 0.7 ^h^	72.6 ± 0.7 ^f^
	1	96.4 ± 0.5 ^o^	89.6 ± 0.6 ^j^	89.5 ± 0.3 ^k^
	Positive control	74.7 ± 0.9 ^gh^	87.6 ± 0.3 ^ij^	89.9 ± 0.1 ^k^
T2	0.25	46.7 ± 0.5 ^b^	47.1 ± 0.4 ^b^	44.8 ± 0.6 ^a^
	0.5	63.8 ± 0.2 ^e^	58.3 ± 0.4 ^d^	56.9 ± 0.3 ^c^
	0.75	78.5 ± 0.7 ^ijk^	70.5 ± 0.5 ^g^	67.8 ± 0.3 ^e^
	1	90.6 ± 0.4 ^n^	86.9 ± 0.3 ^ij^	86.4 ± 0.4 ^ij^
	Positive control	75.7 ± 0.4 ^gh^	86.2 ± 0.4 ^i^	88.1 ± 0.5 ^jk^
T3	0.25	55.7 ± 0.6 ^d^	43.7 ± 0.9 ^a^	45.3 ± 1.1 ^a^
	0.5	73.2 ± 0.7 ^g^	53.1 ± 0.4 ^c^	57.8 ± 0.2 ^c^
	0.75	85.3 ± 0.5 ^m^	66.5 ± 0.4 ^f^	66.7 ± 0.2 ^e^
	1	90.5 ± 0.4 ^n^	77.5 ± 1.5 ^h^	79.4 ± 0.3 ^g^
	Positive control	76.1 ± 0.5 ^ghi^	85.7 ± 0.4 ^i^	87.3 ± 0.3 ^jk^
T4	0.25	47.5 ± 0.5 ^bc^	53.5 ± 0.4 ^c^	51.3 ± 0.5 ^b^
	0.5	55.2 ± 0.5 ^d^	66.6 ± 0.5 ^f^	61.4 ± 0.4 ^d^
	0.75	67.6 ± 0.4 ^f^	76.3 ± 0.8 ^h^	71.4 ± 0.6 ^f^
	1	83.1 ± 0.4 ^lm^	85.2 ± 0.2 ^i^	82.0 ± 0.3 ^gh^
	Positive control	78.9 ± 0.2 ^jk^	84.6 ± 0.3 ^i^	88.1 ± 0.3 ^jk^
T5	0.25	43.3 ± 0.4 ^a^	53.5 ± 0.5 ^c^	51.3 ± 0.3 ^b^
	0.5	53.4 ± 0.5 ^d^	66.5 ± 0.7 ^f^	61.7 ± 0.3 ^d^
	0.75	67.9 ± 0.4 ^f^	75.6 ± 0.6 ^h^	72.3 ± 0.9 ^f^
	1	83.4 ± 0.4 ^m^	86.4 ± 0.5 ^i^	84.1 ± 0.1 ^hi^
	Positive control	77.6 ± 0.3 ^hij^	85.6 ± 0.4 ^i^	87.3 ± 0.3 ^jk^

Values are presented as mean ± SE. Identical lowercase letters within the same column indicate no significant difference in AChE, tyrosinase, α-glucosidase activities, and essential oil application doses, respectively (Tukey test; *p* < 0.05).

**Table 6 plants-14-00116-t006:** Harvesting sites, collecting times, and essential oil yields of studied plants.

Species Code	Species	Harvesting Site	Collection Time	GPS Coordinates	Voucher Specimen	Altitude (m)	Oil Yield (% (*w*/*w*))
T_1_	*T. willdenowii* Boiss.	Lahri (Khenifra)	Jun 2021	32°52′07.5″ N 5°38′10.6″ W	ER-21-55	830	3.7 ± 0.1
T_2_	*T. zygis* subsp. *gracilis*	Ait Ishak (Khenifra)	Jun 2021	32°45′50.4″ N 5°43′59.6″ W	ER-21-60	1223	2.1 ± 0.1
T_3_	*T. broussonnetii* subsp. *Broussonnetii*	Ounagha (Essaouira)	July 2021	31°31′43.7″ N 9°31′26.3″ W	ER-21-57	275	1.4 ± 0.1
T_4_	*T. maroccanus*	Ait Ourir (Marrakech)	July 2021	31°33′57.4″ N 7°37′32.7″ W	ER-21-59	730	1.8 ± 0.1
T_5_	*T. satureioides*	Ijoukak (Marrakech)	Jun 2021	30°59′51.6″ N 8°09′16.2″ W	ER-21-62	2422	1.6 ± 0.1

## Data Availability

The raw data are available from the corresponding authors (Mohamed Ouknin and Lhou Majidi) upon reasonable request.

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
