# Peer review of "Comparative Analysis of Five Moroccan Thyme Species: Insights into Chemical Composition, Antioxidant Potential, Anti-Enzymatic Properties, and Insecticidal Effects"

_plants, 2025, doi:10.3390/plants14010116_

Round 1

Reviewer 1 Report

Comments and Suggestions for Authors

Dear authors of the manuscript,

I find your work very interesting and well writing. I would like to make some suggestions.

First of all, I think that in the introduction part you give so many information about the melon aphlid in comparison to the thyme species that it seems like the main topic of your work is the ecological impact of the insect. I think that you have to give extra information (from the bibliography) about the role of the thyme essential oil or/and the EO of Lamiaceae plant as anti-insecticidal.

Moreover, you should check again the whole manuscript because every species and genus as well as "in vitro" and "in vivo" must be written in italic mode.

In Figure 1 you show the structure of the main compounds which all of them are known. For me is not necessary, but if you want keep them you had better draw them in chemdraw.

In Table 2 you said about Ir J= retention indices on Joulain, but you did not explain in the material and method part what exactly is this indice.

In part 4.5 you did not give any bibliography. Is this method made from the member of your lab or it is performed according to the bibliography?

Author Response

Dear reviewer,

We would like to thank for your very useful comments, which helped to improve the quality of the manuscript.

Answers are highlighted in red color in the manuscript.

Best regards.

Comments

Response

First of all, I think that in the introduction part you give so many information about the melon aphlid in comparison to the thyme species that it seems like the main topic of your work is the ecological impact of the insect. I think that you have to give extra information (from the bibliography) about the role of the thyme essential oil or/and the EO of Lamiaceae plant as anti-insecticidal.

The introduction has been reworded and strengthened as requested.

Moreover, you should check again the whole manuscript because every species and genus as well as "in vitro" and "in vivo" must be written in italic mode.

Each species and genus, as well as the terms “in vitro” and “in vivo”, are written in italics.

In Figure 1 you show the structure of the main compounds which all of them are known. For me is not necessary, but if you want keep them you had better draw them in chemdraw.

In Figure 1, The main compounds are drawn in chemdraw.

In Table 2 you said about Ir J= retention indices on Joulain, but you did not explain in the material and method part what exactly is this indice.

In the materials and methods section, the retention indices on Joulain are explained.

The retention indices (RIs) on Joulain refer to the tabulated retention indices for essential oil components and other volatile compounds, as documented in "The Atlas of Spectral Data of Sesquiterpene Hydrocarbons" by Joulain and König. This reference book provides retention indices primarily obtained on non-polar (e.g., DB-5, HP-5) and polar (e.g., Carbowax, Innowax) columns, aiding in the identification of sesquiterpene hydrocarbons in essential oils.

In part 4.5 you did not give any bibliography. Is this method made from the member of your lab or it is performed according to the bibliography?

In Section 4.5, the methodology is detailed based on the equipment and analytical conditions utilized for all samples processed in our laboratory.

Reviewer 2 Report

Comments and Suggestions for Authors

The manuscript “Comparative Analysis of Five Moroccan Thyme species: Insights into Chemical Composition, Antioxidant Potential, Anti-Enzymatic Properties, and Insecticidal Effects” by Ouknin et al. investigates chemical composition from essential oils (volatiles by GC-MS and micro and macroelements by ICP-AES) from five thyme species derived from Morocco as well as their antioxidant, anti-enzymatic and insecticidal properties.  

The manuscript is well written and under the scope of the journal. It has a clear and sound scientific language, with the good quality scientific English. The methods are adequately chosen and experiments seem to be performed accurately. In my opinion, the manuscript is of good quality, but several issues need to be addressed before the publication.

Major issues

1. The authors claim in the last paragraph of the Introduction that the overreaching goal was to establish correlations between biological activities and the chemical composition of five Moroccan thyme oils whereas no statistical analysis regarding that matter was performed.

2. I am not convinced that the statistical labeling is correctly placed in Tables 3 and 5. The author claims that values marked with different letter indicate significant differences. For example, in Table 3, in column β-Carotene test the sample T5 and BHT are marked as if they are statistically significantly different whereas, by their values, it is highly unlikely that that is true. The same in the Table 5, for example, column AChE (%) sample T2 0.75 and positive control, T3 0.75 and T3 1. Please check.

3. The author should avoid word significant thought the entire manuscript unless they are describing results derived from statistical analysis.

Minor issues

1. The abstract should be shortened following journal guidelines

2. Line 77: EO are NOT typically consisted of phenolics, please correct this statement.

3. Line 111. Elevated consummation of sodium can elevate blood pressure and badly affect cardiovascular system

4. Through the entire manuscript- Latin name of thymus should be written in italic

5. Line 146. The * sign is in the wrong place at the table, pleas correct

5. Line 179.
a) Chemical should be with small letter
b) Several compound names should be corrected: E is written in italic and in brackets, Greek letters are written in italic, compound No. 18. what does THF mean??,  trans- t is small etc.

6. Gallic acid, when not at the beginning of the sentence, is written with small letter g.

7. The author states, in several places in the text and in the conclusion as well, that the thyme species could attribute in human diet whereas it is rich in minerals. Since the thyme is highly aromatic plant and it is not used in large amount, my question to the author would be- in which way should the plant be consumed in order to really attribute to human health in considerable level?

8. Lines 314 and 319. Can the authors give some explanation why is their result different than results from other authors?

9. Materials and method: Section 4.1 should be placed along with other biological activities, after chemical analysis.

Author Response

Dear reviewer,

We would like to thank for your very useful comments, which helped to improve the quality of the manuscript.

Answers are highlighted in red color in the manuscript.

Best regards.

Response to Comments:

Reviewer #2

Comments

Response

1. The authors claim in the last paragraph of the Introduction that the overreaching goal was to establish correlations between biological activities and the chemical composition of five Moroccan thyme oils whereas no statistical analysis regarding that matter was performed.

One of our objectives was to investigate the correlation between the biological activities and the composition of the essential oils of the five thyme species studied. However, the statistical analysis conducted in this context revealed that the synergy between the components of these essential oils does not support such a correlation. Therefore, it would be more appropriate to remove the last sentence in the introduction-objectives section (see manuscript).

2. I am not convinced that the statistical labeling is correctly placed in Tables 3 and 5. The author claims that values marked with different letter indicate significant differences. For example, in Table 3, in column β-Carotene test the sample T5 and BHT are marked as if they are statistically significantly different whereas, by their values, it is highly unlikely that that is true. The same in the Table 5, for example, column AChE (%) sample T2 0.75 and positive control, T3 0.75 and T3 1. Please check.

Statistical analysis is examined and verified for each part (see manuscript).

3. The author should avoid word significant thought the entire manuscript unless they are describing results derived from statistical analysis.

The word significant is avoided throughout the manuscript. However, the statistical analysis conducted in this context revealed that the synergy between the components of these essential oils does not support such a correlation.

1. The abstract should be shortened following journal guidelines.

The abstract is written in accordance with the journal's guidelines.

2. Line 77: EO are NOT typically consisted of phenolics, please correct this statement.

The part has been modified as requested

3. Line 111. Elevated consummation of sodium can elevate blood pressure and badly affect cardiovascular system.

Excessive sodium intake is widely recognized as a significant risk factor for elevated blood pressure, which can lead to severe cardiovascular complications. However, when it comes to aromatic and medicinal plants, their sodium content generally remains within safe and low limits, as indicated by the World Health Organization (WHO). This makes these plants a valuable dietary component, especially for individuals aiming to manage their sodium intake while benefiting from the bioactive compounds they provide. The WHO's findings highlight the potential of these plants to contribute to health and well-being without posing a risk of excessive sodium consumption.

World Health Organization (WHO). Quality Control Methods for Medicinal Plant Materials; WHO: Geneva, Switzerland, 1998.

4. Through the entire manuscript- Latin name of thymus should be written in italic.

All scientific names are written in italics as requested.

5. Line 179.
a) Chemical should be with small letter
b) Several compound names should be corrected: E is written in italic and in brackets, Greek letters are written in italic, compound No. 18. what does THF mean??,  trans- t is small etc.

All chemicals are checked and corrected.

In the context of (Z)-Linalool oxide THF, THF typically refers to the tetrahydrofuran structural motif in the compound. This indicates that the molecule contains a tetrahydrofuran ring—a five-membered cyclic ether with four carbon atoms and one oxygen atom.

6. Gallic acid, when not at the beginning of the sentence, is written with small letter g.

Gallic acid, when not at the beginning of a sentence, is written with the small letter g throughout the manuscript.

7. The author states, in several places in the text and in the conclusion as well, that the thyme species could attribute in human diet whereas it is rich in minerals. Since the thyme is highly aromatic plant and it is not used in large amount, my question to the author would be- in which way should the plant be consumed in order to really attribute to human health in considerable level?

Thyme can contribute to human health through various consumption methods that maximize its mineral content. For instance, thyme can be used in infusions or teas, allowing for the extraction of minerals into a liquid form that can be consumed in larger quantities. It can also be incorporated into functional foods, such as bread, soups, or nutritional bars, to enhance their mineral content. Additionally, thyme used in preservation or pickling processes may enrich preserved foods with its minerals over time. Fresh thyme can be added to salads, smoothies, or as a garnish in larger quantities to increase its dietary contribution. These approaches could make its health benefits more substantial despite its typical use in small amounts.

8. Lines 314 and 319. Can the authors give some explanation why is their result different than results from other authors?

Differences in essential oils composition between studies arise from genetic variation, environmental factors (altitude, soil, climate), geographical and seasonal influences, harvesting practices, and extraction methods. These variables underscore the dynamic nature of EOs composition and the need to account for them in comparative analyses.

9. Materials and method: Section 4.1 should be placed along with other biological activities, after chemical analysis.

Section 4.1 is moved after the chemical analysis as requested.

Reviewer 3 Report

Comments and Suggestions for Authors

In the introduction the beginning part (lines 42-60) should be moved before line 100. I think that the introduction should start with the description of the thyme plant and the information about the essential oils and then add the paragraphs about the studied insects. Therefore the first few lines should contain information about the thyme plant.

Results

Rows 107-129 are not part of the results chapter. They are introductory and refer to the purpose of the paper.

I think they should be rephrased in relation to the results obtained.

Randul 132 all species names should be italicized.

Statistical analysis of the mineral components is missing in order to highlight differences between species (Table 1).

Table 4 attention to the spacing (first column).

In the chapter material and method point 4.2. mention the dates of purchase of the clevenger installation as well as the extraction yield (volume of oil obtained). This point should be detailed.

In point 4.3. mention data on the digester used. For each appliance or installation used, indicate its origin (name, company and country of manufacture, address of the manufacturing company).

Pay attention to the formula, I think it is good to use the formula editor. All the formulas are not clearly shown.

Author Response

Dear reviewer,

We would like to thank for your very useful comments, which helped to improve the quality of the manuscript.

Answers are highlighted in red color in the manuscript.

Best regards.

Response to Comments:

Reviewer #3

Comments

Response

In the introduction, the beginning part (lines 42-60) should be moved before line 100. I think that the introduction should start with the description of the thyme plant and the information about the essential oils and then add the paragraphs about the studied insects. Therefore the first few lines should contain information about the thyme plant.

The introduction has been reworded and strengthened as requested.

Rows 107-129 are not part of the results chapter. They are introductory and refer to the purpose of the paper. I think they should be rephrased in relation to the results obtained.

This section is reformulated according to the results obtained.

Randul 132 all species names should be italicized.

All scientific names are written in italics.

Statistical analysis of the mineral components is missing in order to highlight differences between species (Table 1).

Statistical analysis of mineral components is studied and added in Table 1.

Table 4 attention to the spacing (first column).

In Table 4, the spacing in the first column is checked and corrected.

In the chapter material and method point 4.2. mention the dates of purchase of the clevenger installation as well as the extraction yield (volume of oil obtained). This point should be detailed.

The date of purchase of the Clevenger mill and the extraction yield (volume of oil obtained) are detailed as requested.

In point 4.3. mention data on the digester used. For each appliance or installation used, indicate its origin (name, company and country of manufacture, address of the manufacturing company).

Data on the digester used: its origin (name, company and country of manufacture, address of manufacturing company) are added as requested.

Pay attention to the formula, I think it is good to use the formula editor. All the formulas are not clearly shown.

Formulas are created using the formula editor.

Reviewer 4 Report

Comments and Suggestions for Authors

The text proposal is interesting. Before publishing it, there is work to be done regarding the content and the form, especially concerning the references.

Author Response

Dear reviewer,

We would like to thank for your very useful comments, which helped to improve the quality of the manuscript.

Answers are highlighted in red color in the manuscript.

Best regards.

Response to Comments:

Reviewer #4

Comments

Response

1- A relatively quick and limited search shows that the compositions of some of the Thymus essential oils mentioned in this text have already been the subject of publications. At least a few sentences (including references) should appear in the text at the end of section 2.1. See for example: Food Control, 19(7), 681-687 (2008); Braz. J. Pharmacognosy, 17(4), 477-491 (2007); Ind. Crops Prod., 29, 145-153 (2009) and 143, 111922 (2020); Rec. Nat. Prod., 4(4), 230-237 (2010); Chem. Data Coll., 37, 100797 (2022); ...

Section 2.1. deals only with the compositional results of this work; most of these references are included in the introduction and discussion part of the composition of studied plants.

2- The expression "Surname et al." appears about twenty times. I suggest writing it as follows: "Surname et al.".

The expression “Surname et al.” is written as requested.

3- The identity of various thymes appears many times in the form of "T. saturoides", "T. willdenowii", etc., etc. I suggest using italics "T. saturoides", "T. willdenowii", etc., etc.

All scientific names are written in italics

4- Table 2: In the column header of column 3 - 5, the expression "Ir" appears. This likely refers to the French term "Indice de rétention." I suggest replacing it with the more common English expression RI "Refraction Index."

The expression RI “Refractive index” is modified as requested.

5- Table 2. A fun fantasy of the "Word software". Many readers will probably not see the difference between the dash (-) and the minus sign (–) randomly used in Table 2.

Revised and modified as requested.

6- The letters E and Z and their meanings trans and cis should be in italics: Line 174 and 327 and in Table 2, lines 16, 18 [(Z)-Linalool oxide THF] and 31 (trans-Caryophyllene).

The letters E and Z and their trans and cis meanings are in italics.

References

 1- Some references appear problematic (non-functioning web references, title or list of authors in disagreement, …). See ref: 1; ref. 8; ref. 59; ref. 71; ref. 80.

 2- In some places the issue number is missing: Ref. 4: 45(6); ref. 9: 17(12); ref. 14: 7(1); ref. 15: 22(2); ref. 55: 4(10); ref. 64: 9(6); ref. 80: 69(1); ref. 81: 99(8).

3- Is it an effect of the computer tools used (?), the DOI has not been found. See ref. 5, 6.

 4- There are unnecessary capital letters in the reference titles. See ref. 19; ref. 20; ref. 53.

5- The list of authors is incomplete. See ref.Q 21; ref. 35.

 6- The year of publication of the reference is most often written in bold characters. This is not the case for references 11, 16, 23, 28, 37, 47, 52, 7-

All references are checked and modified according to journal instructions in the case of Journal Articles and Books and Book Chapters, etc.

Others:

a. Ref 25: … Sadler, M.J. Encyclopaedia of food science, food technology and nutrition. Academic press, 1993, pp. 5365. Perhaps it would have been appropriate to specify the relevant part of this encyclopedia.   

 b. Ref. 31: … Rodrigues, M.J.; …

 c. Ref. 40 and 45: The title of the references does not need to be in italics.

 d. Ref. 49: … Kobarfard, F.; …

e. Ref. 61: … Myzus persicae …

f. Ref. 74: … El Messoussi, S., Mounir, A. & Majidi, L…

g. Ref. 77: https://doi.org/10.1016/S0021-9673(01)80947-X h

Ref. 78: … parts of Thymus willdenowii Boiss & Reut. …

i.                                             Ref. 81:

 https://doi.org/10.1002/jps.2600600940

All references are checked and modified according to the journal's instructions.

Round 2

Reviewer 2 Report

Comments and Suggestions for Authors

The authors have addressed all my concerns, and I believe the manuscript is now ready for publication

Author Response

Dear reviewer,

We would like to thank for your very useful comments, which helped to improve the quality of the manuscript.

Answers are highlighted in red color in the manuscript.

Best regards.

Response to Comments:

Significant Figures: The number of decimal places in the uncertainty (±21.45) suggests a higher precision than necessary. Uncertainty is generally reported with one or two significant figures, e.g., 450.6 ± 21 or 451 ± 21. The main value (450.56) should then be rounded to match the precision of the uncertainty.

  • All values in text are revised as requested.

Ensure that all values in the manuscript are reported consistently in terms of significant figures and format.

  • All values in text are revised in terms of significant figures and format as requested.

Ensure that all reported values include appropriate units to avoid ambiguity.

  • All reported values include the appropriate units are verified in the manuscript.

By addressing these points, the manuscript will better align with the conventions of scientific reporting, facilitating reader comprehension and improving the overall quality of the work.

Please, also ensure that the stereochemistry of borneol is visible on the given structure in Figure 1.

  • Please, also ensure that the stereochemistry of borneol is visible on the given structure in Figure 1.

Please, check for errors such as "camphre" in Table 1. It should be "camphor".  Instead of "(E)-Hydrate sabinene", it should be "trans-sabinene hydrate", as it is an accepted trivial name. Instead of "(Z)-Linalool oxide THF", it should be "cis-Linalool oxide THF". Stereodesignators E and Z are only used for the configuration of alkenes and not for cyclic diastereomers.

  • Errors are checked and modified as requested.

Foot note to Table 1, instead of "bRI j = retention indices on the Joulain", it should be "bRI j = retention indices given inJoulain et al.".

  • The footnote to Table 1 is modified as requested.
